# Influence of Gut Microbiota on Metabolism of Bisphenol A, a Major Component of Polycarbonate Plastics

**DOI:** 10.3390/toxics11040340

**Published:** 2023-03-31

**Authors:** Weili Mao, Lingling Mao, Feifei Zhou, Jiafeng Shen, Nan Zhao, Hangbiao Jin, Jun Hu, Zefu Hu

**Affiliations:** 1Department of Pharmacy, Quzhou People’s Hospital, Quzhou Affiliated Hospital of Wenzhou Medical University, Quzhou 310032, China; 2Key Laboratory of Microbial Technology for Industrial Pollution Control of Zhejiang Province, College of Environment, Zhejiang University of Technology, Hangzhou 310032, Chinahujun1988@zjut.edu.cn (J.H.); 3CAS Testing Technical Services Jiaxing Co., Jiaxing 314000, China

**Keywords:** bisphenol A, gut microbiota, bisphenol A glucuronide, bisphenol A sulfate, metabolism

## Abstract

Bisphenol A (BPA) is a major component of polycarbonate plastics and epoxy resins. While many studies have investigated the effect BPA exposure has upon changes in gut microbial communities, the influence of gut microbiota on an organism’s ability to metabolize BPA remains comparatively unexplored. To remedy this, in this study, Sprague Dawley rats were intermittently (i.e., at a 7-day interval) or continuously dosed with 500 μg BPA/kg bw/day for 28 days, via oral gavage. In the rats which underwent the 7-day interval BPA exposure, neither their metabolism of BPA nor their gut microbiota structure changed greatly with dosing time. In contrast, following continuous BPA exposure, the relative level of *Firmicutes* and *Proteobacteria* in the rats’ guts significantly increased, and the alpha diversity of the rats’ gut bacteria was greatly reduced. Meanwhile, the mean proportion of BPA sulfate to total BPA in rat blood was gradually decreased from 30 (on day 1) to 7.4% (by day 28). After 28 days of continuous exposure, the mean proportion of BPA glucuronide to total BPA in the rats’ urine elevated from 70 to 81%, and in the rats’ feces the mean proportion of BPA gradually decreased from 83 to 65%. Under continuous BPA exposure, the abundances of 27, 25, and 24 gut microbial genera were significantly correlated with the proportion of BPA or its metabolites in the rats’ blood, urine, and feces, respectively. Overall, this study principally aimed to demonstrate that continuous BPA exposure disrupted the rats’ gut microbiota communities, which in turn altered the rats’ metabolism of BPA. These findings contribute to the better understanding of the metabolism of BPA in humans.

## 1. Introduction

Bisphenol A (BPA) is an industrial chemical with a high global production volume, being mainly polymerized to produce polycarbonate plastics and epoxy resins [1]. Due to the incomplete polymerization, residual BPA has been detected in numerous plastic consumer products, such as baby bottles, food containers, and beverage cans [2]. This wide-ranging BPA presence, together with the leaching of BPA into foods, renders human BPA exposure ubiquitous [2,3]. For the general population, exposure to BPA predominantly occurs through dietary ingestion, with minor contributions from dermal absorption and dust inhalation [3]. BPA has been detected in the blood, urine, and other bodily fluids of humans from around the world [4,5,6]. For example, 94% of human urine samples (*n* = 294) from across several Asian countries contained measurable levels of BPA, with a concentration range of <0.1–30 ng/mL [7]. Crucially, BPA is biologically active, showing a weak estrogenic activity both in vitro and in vivo [8]. Biochemical assays have demonstrated that BPA has a strong binding affinity with both estrogen receptors *α* and *β*, and also has the potential to disrupt normal hormonal functions in humans [9,10]. Epidemiological data have linked BPA exposure to human diabetes, heart diseases, asthma, and the onset of obesity [11,12,13]. Therefore, human exposure to BPA has become a global concern.

After ingestion, BPA is predominantly metabolized to BPA-glucuronide (BPA-G) and BPA-sulfate (BPA-S) in humans, and then rapidly excreted via urine and feces, with a resulting half-life of <12 h [10]. It has long been considered that BPA-G and BPA-S are biologically inactive, and that neither exhibits obvious in vitro estrogenic activity [14,15]. However, recent studies revealed the endocrine disrupting activities of BPA-G and BPA di-sulfate (BPA-DS) in rat prolactinoma cells, and the cytotoxicity of BPA-G in both human and mice preadipocytes [16,17]. Moreover, BPA-G and BPA-S can be deconjugated to BPA in vivo, mainly in the liver and intestine [18,19], which can act as a potential source of BPA. Following these toxicology studies, some biomonitoring results examined the occurrence of BPA, BPA-G, and BPA-S in humans, measured mainly from serum and urine samples. Volkel, Colnot, Csanady, Filser and Dekant [10] were the first to develop a quantitation method for analyzing BPA-G and BPA in human urine and blood. Later, Liao and Kannan [6] reported higher geometric mean BPA-G concentrations than either BPA or BPA-DS in human urine and serum. Recently, human biomonitoring on maternal and cord serum reported higher BPA-G and BPA-S concentrations than BPA [20]. Therefore, considering the demonstrated toxicities of BPA-G and BPA-S, it is necessary that we understand the mechanisms that underlie the metabolism of BPA into its conjugated metabolites if we are to accurately and fully understand the toxic effects of BPA on humans.

Gut microbiota are a large and complex composition of microbes in the gastrointestinal tract [21], which play an important role in metabolic processing, energy homeostasis, and nutrient absorption in humans [22]. Changes in the gut microbiome composition have been linked to the development of many human diseases [23]. The composition of gut microbiota is highly sensitive to exogenous stressors and environmental contaminants [24]. Many studies have demonstrated that exposure to estrogenic compounds, including BPA, could remarkably change the composition of gut microbiota in organisms [22]. For example, Lai et al. reported that dietary exposure to BPA greatly altered the community of mouse gut microbiota, enhancing the growth of *Proteobacteria*, a biomarker of intestinal microbial dysbiosis [25]. Despite the liver perhaps being better known for it, gut microbiota are also involved in the metabolism of a wide variety of xenobiotics (i.e., drugs and heavy metals) in humans, and could thus significantly regulate the toxic effects of xenobiotics [26]. Using an in vitro human intestinal microbial ecosystem, Wang, Rui, Nie and Lu [27] found that the bioavailability of BPA declined over the course of the gastrointestinal digestion process. However, the influence of gut microbiota changes on the metabolism of BPA into BPA-G and BPA-S in humans remains insufficiently understood.

In this study, Sprague Dawley (SD) rats were intermittently (i.e., at a 7-day interval) or continuously dosed with 500 μg BPA/kg bw/day for 28 days, by means of oral gavage. SD rats are a reliable model animal that have been widely used to understand the toxicokinetic behaviors of BPA in humans. Our BPA dosing level was chosen to render rat blood BPA levels similar to those in the general human population. Then, we identified the concentrations of BPA and its major conjugated metabolites in the rat’s feces, urine, and blood across both dosing timescales. The associated change in gut microbiome community structure in conjunction with each dosing interval was also investigated using 16S rRNA amplicon sequencing. Finally, the influence of the rats’ gut microbiota change on their metabolism of BPA was explored by analyzing the correlations between the gut microbiome abundance and the proportion of BPA and its metabolites in the rats’ feces, urine, and blood. This study aims to contribute to a better understanding of the metabolism of BPA in humans.

## 2. Materials and Methods

**Standards and Reagents.** BPA (2,2-(4,4′-dihydroxydiphenyl)propane; purity > 98%), bisphenol A mono-*β*-d-glucuronide (BPA-G; >98%), bisphenol A mono-sulfate (BPA-S; >98%), bisphenol A bis-*β*-d-glucuronide (BPA-BG; >97%), bisphenol A di-sulfate (BPA-DS; >97%), ^13^C_12_-BPA, ^13^C_12_-BPA *β*-d-glucuronide (^13^C_12_-BPA-G; >98%), and D_6_-BPA mono-sulfate (D_6_-BPA-S; >98%) were purchased from Toronto Research Chemicals (North York, ON, Canada). Dimethyl sulfoxide (DMSO), methanol, physiological saline (0.9% NaCl solution), pure water, and ammonium acetate were obtained from Sigma-Aldrich (Oakville, ON, Canada).

**Animal and Experimental Design.** Sprague Dawley rats (ten weeks old, body weight 230–280 g) were obtained from SLRC Animal Laboratory (Shanghai, China). All animals were housed (5 rats/cage) in a temperature (22–25 °C), humidity (40–50%), and light (12/12 h light/dark cycle) controlled house under specific pathogen-free conditions, in addition to being given ad libitum access to standard food (Medicience Ltd., Jiangsu, China) and water. Before conducting the experiment, all animals were allowed to acclimate for seven days. All SD rats were humanely treated throughout the experiment, following protocols approved by the Zhejiang University of Technology’s Animal Ethics Committee.

An initial experiment was conducted on 10 SD rats to investigate whether the metabolism of BPA was changed under stable gut microbiota community. On days 1, 8, 15, 22, and 29 (7-day interval), all SD rats were administered with BPA (reconstituted in 50% DMSO/water) once only via gavage at a dose of 500 μg/kg bw. After oral BPA intake, the SD rats (*n* = 10) were placed into individual metabolic cages, and the total resulting rat feces and urine samples from natural micturition were collected for the 24 h that followed BPA administration. After 24 h had elapsed from the administration of BPA, whole blood samples (1 mL) were collected from each rat’s vena caudalis.

A second experiment was performed on another 10 SD rats to assess whether the altered gut microbiota could change their metabolism of BPA. All SD rats were continuously administered with BPA (reconstituted in 50% DMSO/water) at 500 μg/kg bw using oral gavage once a day for 29 days. After being dosed on days 0, 1, 3, 5, 7, 9, 15, and 29, the BPA-exposed rats (*n* = 10) were placed in individual metabolic cages for 24 h, to collect urine and feces samples. After 24 h had elapsed from the administration of BPA, whole blood samples (1 mL) were collected from each rat’s vena caudalis.

In both experiments, SD rats (*n* = 10) were set as control groups, dosed with the same amount of 50% DMSO/water and given ad libitum access to clean water and food. The total volume and weight of the urine and feces collected from individual rats was accurately measured. The rats’ blood was collected in BD Vacutainer^®^ tubes (embedded with sodium heparin; NJ, USA). Dry ice was placed so as to surround the urine and feces collection glass vessels during the sample collection period. After collection, the rat feces samples were transferred to RNase-free microfuge tubes (15 mL; Conical, Thermo Fisher; Markham, ON, Canada), and stored at –80 °C until extraction and analysis.

**Sample Extraction.** Prior to extraction, all samples were spiked with ^13^C_12_-BPA, ^13^C_12_-BPA-G, and D_6_-BPA-S (5 ng each), working as internal standards. Rat urine samples were diluted with pure water, and then extracted using solid-phase extraction (SPE), following the method of Liao and Kannan [6]. The rat blood samples (500 μL) were directly extracted with methanol. Feces samples were freeze-dried, homogenized, and then extracted using 80% methanol/water, with additional purification using Supelco Envi-Carb cartridges (Sigma-Aldrich; Oakville, ON, Canada). In addition, spike and recovery experiments were conducted to evaluate the extraction efficiency of analytes. Detailed extraction procedures and recovery experiments are provided in the Appendix A.

**Instrumental Analysis.** All sample extracts were analyzed using ACQUITY liquid chromatography, coupled with a triple quadrupole mass spectrometer (XEVO_TQS; Waters; Milford, MA, USA) [28,29]. The liquid chromatography was performed using an HSS T_3_ column (1.8 μm, 2.1 × 50 mm; Waters, MA, USA), with the mobile phase composed of methanol and water (2.0 mM ammonium acetate, pH = 7). The column temperature and flow rate of mobile phase were maintained at 40 °C and 0.2 mL/min, respectively. The gradient elution was initially held at 20% methanol for 0.5 min, increased to 40% methanol over 1.0 min, and ramped to 95% methanol over 6.0 min, which was then held for 2 min, and finally returned to the starting conditions. The mass spectrometer was operated in negative ionization mode, and spectral data were record using multiple reaction monitoring (MRM; 2 transitions per analyte). Detailed MRM transitions of target analytes are provided in Appendix A.

**QA/QC.** Pure methanol (10 μL) was analyzed between every ten samples to monitor carryover contamination, and no obvious cross contamination between injections was found. Several steps were taken to achieve a very low to undetectable level of background BPA pollution. Despite fresh glassware being used in the every extraction procedure, the procedural blanks that were analyzed along with every ten real samples were still found to consistently contain low concentrations of BPA, which possibly originated from SPE cartridges. BPA and BPA conjugates in the collected samples were quantified using an internal calibration method. Background BPA concentrations were subtracted from the quantified BPA concentrations of the real rat samples. Limits of detection (LODs) were defined as the concentrations of analytes which corresponded to a signal-to-noise ratio of 3 in sample extracts from control rats; these were found to be in the range of 0.047–0.088 ng/mL, 0.039–0.077 ng/g, and 0.039–0.15 ng/mL in the rats’ blood, feces, and urine, respectively. Extraction recoveries of target analytes in rats’ blood, feces, and urine ranged from 72 to 114%. Detailed LODs and extraction recovery of analytes are shown in Appendix A, respectively.

To avoid the in-source fragmentation of BPA conjugates, the capillary voltage of the ion source was set at a low level (–1.0 kV), resulting in a slight sacrifice to detection sensitivity. Fragmentation of BPA conjugates in the second quadrupole can generate BPA, which can greatly increase the quantified BPA concentrations. This interference was minimized with the baseline separation of BPA and its conjugated metabolites. A T3 column, having a strong retention capacity with hydrophilic compounds, was used to reduce both the serous tailing and the shifting retention time of the peak of BPA-DG. Typical chromatograms and molecular structure of target analytes in both the standard solution and rat blood are shown in Figure 1, and again in the Appendix A. Given the instability of BPA conjugates [30], special care was taken to avoid deconjugation during sample collection and extraction. Those blood, feces, and urine samples which had been spiked with ^13^C_12_-BPA-G and D_6_-BPA-S (at 10 or 100 ng/mL) were analyzed alongside of the real samples. No measurable ^13^C_12_-BPA or D_6_-BPA was detected in these fortified samples, demonstrating a negligible deconjugation of BPA metabolites in the sample analysis process. BPA metabolites were not detected in any control rat blood, urine, or feces sample. In control rats, BPA alone was detected in feces at levels around the LOD (a mean of 0.07 ng/g); this was accordingly subtracted from BPA concentrations in the feces of exposed rats.

**Fecal Bacterial DNA Extraction and 16S rRNA Sequencing.** Bacterial DNA was extracted from rat fecal samples using an E.Z.N.A.^®^ Stool DNA Kit (50T, Omega Bio-Tek; Norcross, GA, USA), in accordance with the manufacturer’s protocol. The purity of extracted DNA was determined with a NanoDrop 2000 (Thermo Scientific; Waltham, MA, USA). After that, the DNA was PCR-amplified with barcoded primers (27F: 5′-AGRGTTYGATYMTGGCTCAG-3′ and 1492R: 5′-RGYTACCTTGTTACGACTT-3′), targeting the V1−V9 regions of the bacterial 16S rRNA gene. The PCR reaction mixture (20 μL; performed in triplicate) included 4 μL of 5 × FastPfu buffer, 2 μL of 2.5 mM dNTPs, 0.8 μL of each primer (5 μM), 0.4 μL of FastPfu polymerase, and 20 ng of extracted DNA. The PCR (TransGen AP221-02: TransStart Fastpfu DNA Polymerase) program began with a denaturation step (94 °C; 3 min), followed by denaturation (20 cycles of 1 min; 94 °C), annealing (1 min; 65 °C to 57 °C with a 1°C reduction every two cycles—one cycle at 56 °C, and one cycle at 55 °C), elongation steps (72 °C; 1 min), and a final 6 min extension at 72 °C. The PCR products were pooled, purified with the MiniBest DNA Fragment Purification Kit Ver 4.0 (TaKaRa; Tokyo, Japan), and then sequenced using an Illumina Miseq (Illumina; San Diego, CA, USA). Prior to sequencing, the DNA library was quantified using Library Quantification Kits (KAPA Biosystems; Merck, Rahway, NJ, USA).

**Sequencing Data Analysis.** The obtained raw FASTQ files were demultiplexed (using 8-bp barcodes) and quality-filtered using the QIIME software [31]. Readings with any unknown bases, >2 mismatches to the primers, >1 one mismatch to the barcode, or a <50 bp length were discarded. Those DNA readings with 3 consecutive low-quality bases calls were truncated. After that, overlapped paired-end readings were merged to tags (USEARCH), which were further stepwise clustered to Operational Taxonomic Unit (OTU) with a ≥97% sequence similarity (UPARSE; http://drive5.com/uparse/, accessed on 5 March 2023). A representative sequence from each OTU was taxonomically classified with a confidence threshold of 80%, using the Ribosomal Database Project (RDP) classifier (http://rdp.cme.msu.edu/, accessed on 5 March 2023) against the SILVA database (http://www.arb-silva.de, accessed on 5 March 2023). In this study, taxonomical classification was primarily focused on gut microbiota at phylum and genus levels. Unknown, unclassified, and unassigned classifications were omitted from the final dataset.

## 3. Results

The oral BPA dose is much lower than its reported lowest observed adverse effect level (200 mg/kg bw/day) for rodent [32]. In the experiment process, no obvious diarrhea, dehydration, or vomiting were observed in any SD rats. In control rats, among target analytes, only BPA was detected in rat feces, with concentrations of <0.l ng/g.

**BPA-Induced Gut Microbiome Change.** Under BPA exposure at 7-day intervals, the rats’ gut microbial communities were stable for 29 days, and *Firmicutes* (mean 52–55%) and *Bacteroidetes* (31–35%) were consistently the predominant phyla in rat gut, followed by *Proteobacteria* (7.7–8.6%) and *Actinobacteria* (3.5–4.7%; Figure 2a). This profile is consistent with the human gut, in which *Firmicutes* is always the major phylum [33]. At the genus level, the rats’ gut microbial communities were also consistent, with *Lactobacillus* (mean 36–39%), *Muribaculaceae* spp. (12–16%), and *Bacteroides* (11–14%) being the dominant genera (Figure 2b). Moreover, the use of alpha diversity analysis (Appendix A), a Chao1 estimator (mean 474–480), and the Shannon index (5.5–5.7) also suggest that the diversity of the gut bacterial community was conserved under BPA exposure at 7-day intervals. Overall, these results suggested that BPA exposure at 7-day intervals did not significantly change the gut microbiome of rats over the course of 29 days.

After 29 days of continuous oral BPA administration, the rats’ gut microbial community structures had been significantly changed (Figure 2c,d). At the phylum level, the relative abundances of *Firmicutes* and *Proteobacteria* were significantly (*p* < 0.05) increased, while those of *Bacteroidetes* and *Actinobacteria* had significantly (*p* < 0.05) declined. The continuous oral BPA exposure also greatly reduced the alpha diversity of the rats’ gut bacteria, decreasing from 464 (on day 1) to 342 (by day 29), in addition to decreasing the mean Chao1 and Shannon index, which fell from 5.6 to 4.1, respectively (Appendix A). Principal coordinate analysis (PCoA) also suggested that the rats’ gut microbial community structures between day 1 and 28 showed significant differences, as evaluated at the genus level (PERMANOVA; *p* = 0.003; Appendix A). PCoA results showed that rats on day 1 and 29 were well separated, with 73% and 9.3% of the variation explained by principal component (PC) 1 and PC2, respectively.

**BPA Metabolite Profile in Rat Blood.** Under the 7-day interval BPA exposure, concentrations of BPA and its conjugated metabolites in the rats’ blood did not change significantly with each dosing time (ANOVA, *p* = 0.15–0.33; Figure 3a). In the rats’ blood, the majority (mean 94–95%) of BPA (i.e., the total molar concentrations of BPA and its conjugated metabolites) was present in conjugated forms, with BPA-G (accounting for mean 73–76% of total BPA) being more abundant (*p* < 0.01) than BPA-S (constituting a mean 25–28% of the total BPA) (Figure 4a).

Following 29 days of exposure, remarkable changes in the profiles of BPA-S and BPA-G were observed in rat blood (Figure 4b). For example, the mean proportion of BPA-S to total BPA in the rats’ blood gradually decreased from 30 (on day 1) to 7.4% (by day 29; *p* < 0.01), while that of BPA-G was elevated from 71 (on day 1) to 85% (by day 29; *p* = 0.021). The proportion of BPA in rats’ blood slightly increased with each instance of exposure, from 6.4 (on day 1) to 8.5% (by day 29; *p* < 0.01). After 9 days of continuous BPA intake, the rats’ blood concentrations of BPA and its metabolites became stable, as evidenced by their being no significant (ANOVA, *p* = 0.35–0.40; Figure 3b) differences in their concentrations between days 9, 15, and 29. A previous study on female mice that had been subjected to repeated oral BPA dosing for 28 days also did not find significant accumulation of BPA in the mice’s serum [34].

**BPA Metabolite Profile in Rat Excreta.** The total amount of BPA excreted in rat urine and feces within 24 h of exposure was calculated, based on the molar concentrations of BPA and its metabolites (Appendix A). Under the 7-day interval BPA exposure regimen, the relative amount of BPA excreted in rat urine and feces stayed stable, with a mean 27–32% and 68–73% of BPA excreted in rat urine and feces, respectively. In rat urine specifically, BPA was mainly present in conjugated forms, with BPA in glucuronide form accounting for a mean 74–77% of total excreted BPA. Contrastingly, in rat feces the majority (78–81%) of excreted BPA was present in its free form, and the remaining BPA being mainly in the sulfate form.

The profile of BPA and its metabolites in rat excreta contrasted greatly for the continuous exposure regimen. For example, the mean proportion of BPA-G in the rats’ urine elevated from 70 (on day 1) to 81% (by day 29), while that of BPA-S declined from 24 (on day 1) to 8.7% (by day 29). In the rats’ feces, the mean proportion of BPA gradually decreased from 83 (on day 1) to 65% (by day 29). Beyond nine days of continuous exposure, the total BPA recovered in the rats’ urine and feces did not differ greatly. Based on the data on days 9, 15, and 29, the 24 h cumulative excretion of BPA via urine and feces accounted for 21–28% (mean 25%) and 72–79% (76%) of total excreted BPA, respectively (Appendix A).

**Association Between Gut Microbiome and Metabolism of BPA.** To explore which kinds of gut microbiota were associated with the metabolism of BPA in rats, a correlation analysis was performed between the abundance of gut microbiota at the genus level and the molar proportion of BPA metabolites to total BPA (Figure 5). The 7-day interval BPA exposure did not cause a significant perturbation to the rats’ gut microbiome, nor to the profiles of BPA metabolites between the rats’ blood and excreta over 29 days. Despite this, significantly correlations were still occasionally found between the abundances of gut microbiota and the molar proportion of BPA in rat urine and blood. For example, greater *Negativibacillus* (Spearman’s correlation coefficient, *r*_s_ = 0.41, *p* = 0.022) and *Parabacteroides* (*r*_s_ = 0.33, *p* = 0.035) abundances were significantly associated with a higher proportion of BPA in the rats’ blood. The abundances of *Romboutsia* (*r*_s_ = 0.43, *p* = 0.024) and *Turicibacter* (*r*_s_ = 0.45, *p* = 0.016) were negatively correlated with the proportion of BPA in the rats’ urine. In summary, under a 7-day interval BPA exposure, 5 out of 32 identified gut microbial genera were associated with the proportion of BPA and its metabolites in the rats’ urine or blood.

Under continuous BPA exposure, many significant correlations were exhibited between the perturbed gut microbial genera and the altered profiles of BPA metabolites in the rats. For example, abundances of 27, 25, and 24 gut microbial genera were significantly correlated with the proportion of BPA or its metabolites in rat blood, urine, and feces, respectively. Comparatively, fewer kinds of gut microbiota were correlated with the proportion of BPA-BG and BPA-DS in the rats, relative to BPA-G and BPA-S. For instance, only four species of gut microbiota at the genus level (*Shuttleworthia*, *Lactococcus*, *Ruminococcaceae UCG-013*, and *Negativibacillus*) were correlated with the proportion of BPA-DS in the rats’ blood.

Overall, these results showed that continuous BPA exposure induced a significant perturbation in the rats’ gut microbiota, which in turn substantially altered the metabolism of BPA in the rats, as evidenced by gut microflora-related changes in the BPA metabolite profiles of the rats’ blood and excreta.

## 4. Discussion

Diet intake is considered the major source of human exposure to BPA [35]. The oral administration adopted in this study is very relevant to the actual human BPA exposure scenario. Under continuous dosing at 500 μg BPA/kg bw/day, rat blood BPA concentrations did not accumulate significantly over time, which is mainly due to the rapid elimination of BPA [36]. Rat blood BPA concentrations under both intermittent and continuous exposure (mean < 2.0 ng/mL) are comparable to the reported human blood BPA levels (mean < 0.1–2.5 ng/mL) [5,37].

Under the 7-day interval BPA administration regimen, BPA in glucuronide form comprised the major fraction of the total BPA in the rats’ blood, followed by minor BPA-S. This is consistent with the in vivo results in monkeys and humans [38,39,40]. Draganov, Markham, Beyer, Waechter, Dimond, Budinsky, Shiotsuka, Snyder, Ehman and Hentges [41] had reported that BPA was primarily present in the glucuronide form in mice blood following oral dosing, and BPA-S only contributed 0.46–4.8% of the total BPA metabolites. The predominance of BPA-G in the rats’ blood is mainly attributed to the strong first-pass metabolism that occurs in the liver and intestines [41]. However, two biomonitoring studies have reported markedly higher BPA-S concentrations than BPA-G in human maternal and cord serum, with the authors proposing that this was due to the back transfer of BPA-S in cord blood to maternal serum [20,42].

Under the 7-day interval exposure regimen, BPA was predominantly excreted in the rats’ feces after oral intake. Previous studies on rats had consistently pointed to fecal excretion as the major elimination route for BPA [43,44,45]. For example, Pottenger, Domoradzki, Markham, Hansen, Cagen and Waechter [45] reported that rats excreted approximately 80% of total BPA via feces within 24 h of single oral dosing. In the rats’ feces, most BPA was primary present in the free form (accounting for a mean 78–81% of total BPA), and the remaining BPA was present in the BPA-S form. BPA-G only accounted for <1% of the total BPA in the rats’ feces. These results show some consistency with previous works, which have consistently reported free-form BPA as being predominant in the rats’ feces after oral BPA dosing [43,44,46]. This may be due to enterohepatic circulation, which transfers BPA-G from the liver to the intestine via bile secretion [47], there to be hydrolyzed to BPA by gut microflora [44,46], resulting in elevated fecal excretion of free-form BPA. Due to the expected absence of enterohepatic circulation of BPA in humans [10], the proportion of free-form BPA may be lower in human feces than in rat feces. However, we observed a distinct profile of BPA metabolites in the rats’ urine, relative to the rats’ feces, with BPA-G being much more abundant than BPA or BPA-S. This finding is consistent with previous outcomes [43,44,45]. For example, urinary BPA metabolites in rats following an oral BPA administration also showed the predominance of BPA-G (84% of total BPA), compared with BPA (6.1%) and BPA-S (3.1%) [45]. The rats’ urinary profiles for BPA metabolites were similar to those reported in human urine from America [6,48,49]. The high hydrophilicity of BPA-G may contribute to its being excreted predominantly via urine, and also to its correspondingly reduced levels in the rats’ feces.

After 29 days of continuous BPA exposure, the abundances of *Firmicutes* and *Proteobacteria* were greatly increased, while those of *Bacteroidetes* and *Actinobacteria* had significantly declined. Previous studies have demonstrated that BPA exposure greatly perturbs the gut microbial structure of organisms at the phylum level. Similarly, Feng et al. found that oral BPA exposure (equivalent to 50 μg BPA/kg bw/day) significantly increased (*p* < 0.05) the abundance of *Proteobacteria* in their mice’s guts [50]. The perinatal exposure of rabbits to BPA (200 μg/kg bw) reduced the abundance of *Bacteroidetes* in the gut of their male offspring [51]. It has been reported that a single-dose of BPA exposure could elevate the abundance of *Firmicutes* in zebrafish intestines [24]. We observed that the alpha diversity of our rats’ gut bacteria was also greatly reduced, consistent with previous studies [52]. For example, developmental BPA exposure reduced the diversity of gut microbiota composition in both mice and rabbits [51,53]. The underlying mechanism through which BPA alters the gut microbiome remains unclear. Some studies have proposed that BPA may exert effects on the gut microbiota community through an estrogenic mechanism [53].

Comparison of intermittent and continuous BPA exposure suggests that changes in the rats’ gut microbiome induced by a continuous BPA exposure may in turn have altered the metabolism of BPA in the rats. At the phylum level, the abundances of *Firmicutes* and *Proteobacteria* were positively correlated with the proportion of BPA-G in the rats’ blood. This partially explains the higher abundance of BPA-G in the rats’ urine, since BPA-G was mainly excreted through urine. *Firmicutes* have been shown to have an obviously positive correlation with the BPA degradation in Gonghu Bay sediment, and likewise that high concentrations of BPA can accelerate the BPA removal ascribable to *Firmicutes* [54]. This suggests that *Firmicutes* may have the ability to biodegrade BPA. Likewise, Fernandez, Reina-Perez, Astorga, Rodriguez-Carrillo, Plaza-Diaz and Fontana [52] have reported that many *β*-glucuronidase bacteria in human feces belong to the *Firmicutes* phylum. The abundance of *Bacteroidetes* was negatively correlated with the proportion of BPA in the rats’ blood. Koestel, Backus, Tsuruta, Spollen, Johnson, Javurek, Ellersieck, Wiedmeyer, Kannan, Xue, Bivens, Givan and Rosenfeld [55] have reported that *Bacteroidetes*, capable of degrading bisphenols, were negatively associated with serum BPA concentrations in dogs, after an oral dietary BPA intake.

One obvious change in the rats’ gut microbiota, at the genus level, was the increase in the level of *Blautia* and decrease in the level of *Lactobacillus*. Early life BDE-47 exposure also increased the abundance of the genus *Blautia* in adult male mice [56]. In Cd-treated (5 μg/L) zebrafish, the abundance of *Blautia* was more abundant than the control zebrafish [57]. *Lactobacillus* has been reported to be tolerant or resistant to BPA biodegradation [58], and this may have led to its decrease in abundance in our rats’ gut after BPA intake. We also observed that the relative abundances of *Escherichia-Shigella*, *Ruminococcaceae UCG-005*, *Marvinbryantia*, and *Lachnoclostridium* were also increased in the rats’ gut. Both *Escherichia-Shigella* and *Ruminococcaceae* bacterial groups possess *β*-glucuronidase enzymes [15,40]. This partially explains the increased proportion of BPA-G in the rats’ blood that showed a greater abundance of *Escherichia-Shigella* and *Ruminococcaceae UCG-005*. In contrast, we found that the abundance of *Muribaculaceae-norank* decreased after continuous BPA exposure. The down-regulation of *Muribaculaceae* in gut of adult mice caused by a low PCB dose has been reported [59]. Notably, fewer kinds of gut microbiota were correlated with the proportion of BPA-BG and BPA-DS in rats, relative to BPA-G and BPA-S. This may be because the conjugation of BPA to BPA-BG and BPA-DS is primarily driven by enzymes in the rats’ liver.

**Significance, Limitation, and Perspective.** Despite BPA being rapidly metabolized to BPA conjugates in humans and then rapidly excreted, this does not necessary mean there is a negligible risk of BPA exposure, since BPA metabolites are still widely present in human blood [4]. Moreover, BPA metabolites have been proven to have toxic effects on humans [4,9,10]. Therefore, it is important to understand the underlying mechanisms responsible for the metabolism of BPA in humans. This study has principally aimed to demonstrate the way in which continuous BPA exposure disrupted the rats’ gut microbiota communities, which in turn altered the metabolism of BPA in the rats. The specific gut microbiota that most significantly correlated with the profile of BPA metabolites in the rats were also identified. These findings contribute to the better understanding of the metabolism of BPA in humans. Notably, gut microbiota is highly variable between different species. Thus, the transposition of the results of this study to humans should be performed with caution. More studies are needed to further evaluate the relative contribution of gut microbiota to the occurrence of BPA metabolites in humans.

## Figures and Tables

**Figure 1 toxics-11-00340-f001:**
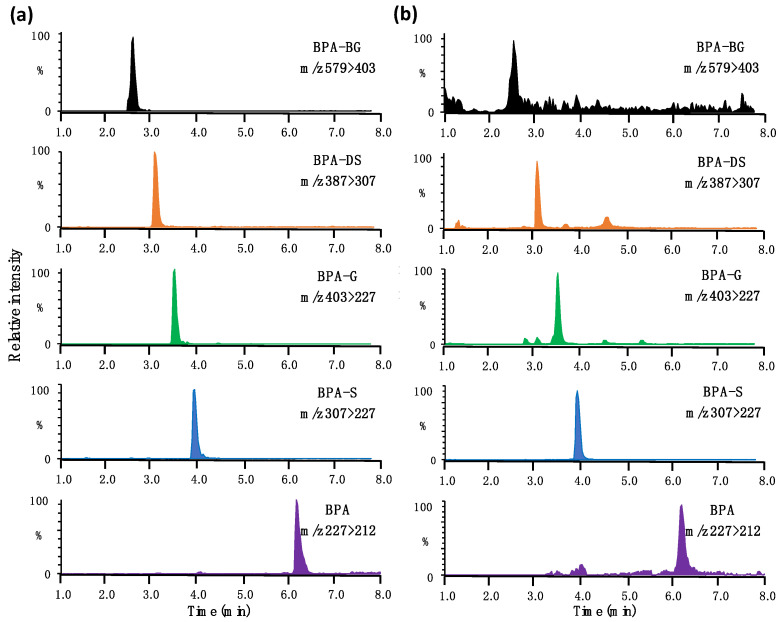
Representative chromatograms of BPA and its conjugated metabolites in (**a**) the 50% methanol/water solution and (**b**) rat blood.

**Figure 2 toxics-11-00340-f002:**
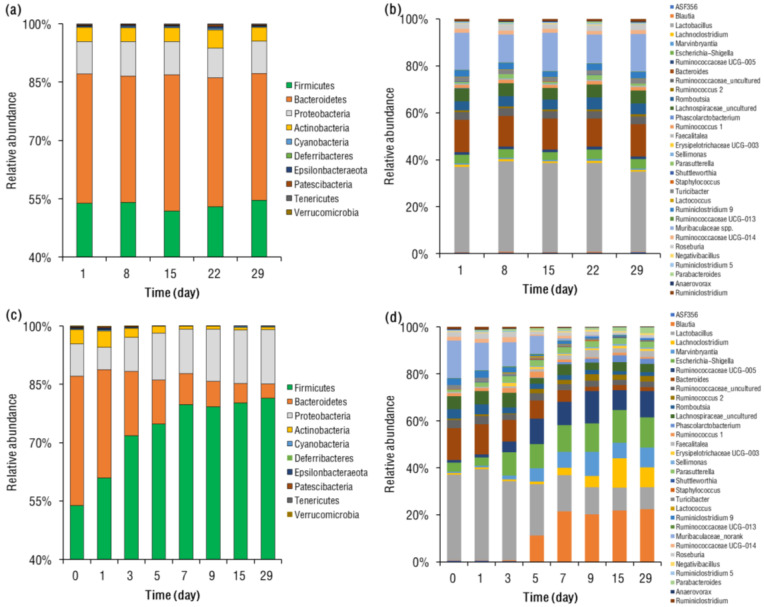
Change in the relative abundances of gut microbiota at (**a**) phylum and (**b**) genus levels in SD rats over time, following intermittent oral administration of BPA. Change in the relative abundances of gut microbiota at (**c**) phylum and (**d**) genus levels in SD rats over time, following continuous oral administration of BPA.

**Figure 3 toxics-11-00340-f003:**
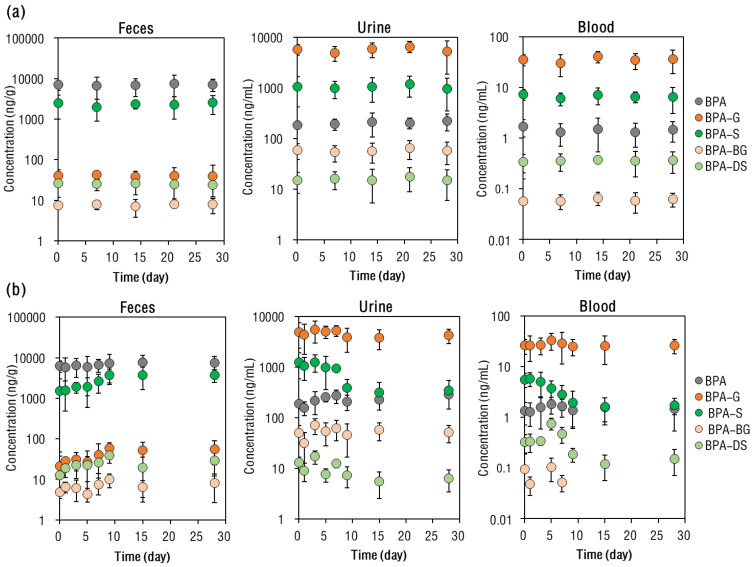
Mean concentrations of BPA and its conjugated metabolites in SD rat feces, urine, and blood, following (**a**) intervallic and (**b**) continuous oral BPA administration. The vertical bars represent standard deviation.

**Figure 4 toxics-11-00340-f004:**
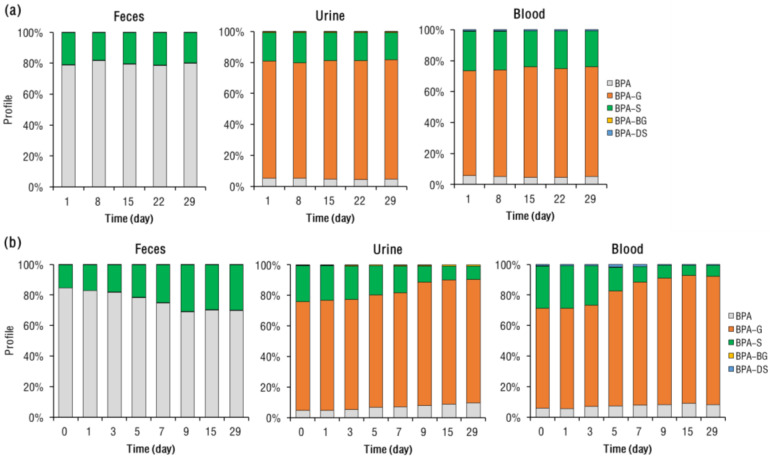
Change in the molar concentration profile of BPA and its conjugated metabolites in feces, urine, and blood of SD rats over time, following (**a**) intervallic and (**b**) continuous oral BPA exposure.

**Figure 5 toxics-11-00340-f005:**
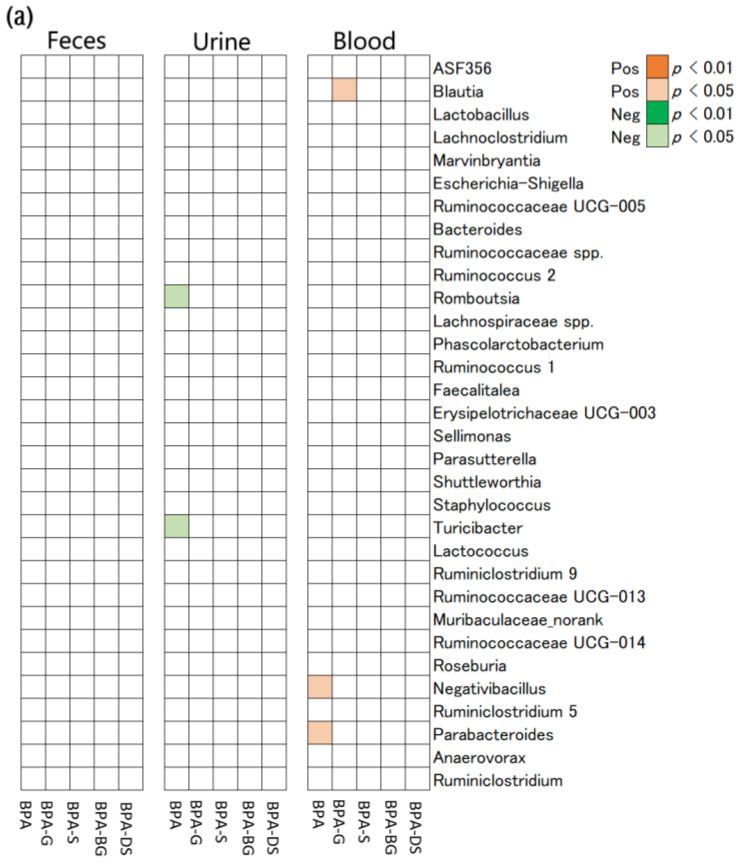
Correlations between the abundances of rat gut bacteria at the genus level and proportion of BPA and its metabolites in feces, urine, and blood of rats following (**a**) intervallic and (**b**) continuous BPA exposure. The red and green squares indicate a significantly positive and negative correlation, respectively.

## Data Availability

Not applicable.

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
