# Peer review of "Influence of Gut Microbiota on Metabolism of Bisphenol A, a Major Component of Polycarbonate Plastics"

_toxics, 2023, doi:10.3390/toxics11040340_

Round 1
Reviewer 1 Report
The authors carried out well-planned studies and applied appropriate research methods. They provided a detailed description of the experiments. The method of BPA administration (intermittent or continuous) was observed to correlate with changes in the amount of specific microorganisms in the intestines, and consequently with changes in the concentration of BPA and its metabolites in excretions and blood. The thesis contained in the title of the article has been indirectly confirmed. The test results are certainly important from the point of view of toxicology.
Author Response
Response: we thank the reviewer very much for your comments.

Reviewer 2 Report
In this article, the authors present a study on the influence of BPA on the gut microbiota. Overall, the study is very well done, with a very well structured and robust experimental part. The analytical chemistry elements are covered in an extremely well-paced manner. I have no revisions for what concerns the methodological part.
I suggest some minor revisions:
The introduction should contain more international references, with studies done in other countries as well. The references are too much oriented on Asian studies, while the issue and relevance of the study is clearly international. It is also important to introduce a reflection on the relevance of the study: it is well known that the gut microbiota is highly variable from species to species, even from individual to individual. Thus, this study has great methodological potential and as a proof of principle, but its conclusions refer to Sprague-Dawley rats and the transposition of the results to other species, particularly humans, must be contextualized and relativized (*).
Figure 1 should, in my opinion, be moved to the IS.
Figure 5b and d is unclear, I would suggest proposing two histograms, one with the most frequent species and one with the least frequent. From this figure it is difficult to distinguish certain species.
The discussion needs to be rewritten; there are too many technical elements proper to the results. The results, i.e., quantitative data, species lists, factual observations, need to be integrated into the results part. The discussion should focus on the study as a whole, highlighting limitations and qualities, and discussing perspectives. It is also important to integrate here the elements of significance of the study as highlighted above (*).
Author Response
In this article, the authors present a study on the influence of BPA on the gut microbiota. Overall, the study is very well done, with a very well structured and robust experimental part. The analytical chemistry elements are covered in an extremely well-paced manner. I have no revisions for what concerns the methodological part.
I suggest some minor revisions:
Response: we thank the reviewer very much for your comments.
The introduction should contain more international references, with studies done in other countries as well. The references are too much oriented on Asian studies, while the issue and relevance of the study is clearly international. It is also important to introduce a reflection on the relevance of the study: it is well known that the gut microbiota is highly variable from species to species, even from individual to individual. Thus, this study has great methodological potential and as a proof of principle, but its conclusions refer to Sprague-Dawley rats and the transposition of the results to other species, particularly humans, must be contextualized and relativized (*).
Response: as suggested, we have added some international references in the introduction section (Azzouz, A., Rascon, A.J. and Ballesteros, E. (2016) Simultaneous determination of parabens, alkylphenols, phenylphenols, bisphenol A and triclosan in human urine, blood and breast milk by continuous solid-phase extraction and gas chromatography-mass spectrometry. J Pharm Biomed Anal 119, 16-26. Ong, H.-T., Samsudin, H. and Soto-Valdez, H. (2022) Migration of endocrine-disrupting chemicals into food from plastic packaging materials: an overview of chemical risk assessment, techniques to monitor migration, and international regulations. Critical reviews in food science nutrition 62(4), 957-979.). We also indicated in the manuscript that the transposition of the results of this study to humans should be performed with caution: “Notably, gut microbiota is highly variable between different species. Thus, the transposition of the results of this study to humans should be performed with caution.”
Figure 1 should, in my opinion, be moved to the IS.
Response: since analyzing methods for BPA and its conjugated metabolites are vital, so we kept Figure 1 in the manuscript.
Figure 5b and d is unclear, I would suggest proposing two histograms, one with the most frequent species and one with the least frequent. From this figure it is difficult to distinguish certain species.
Response: we have revised that, as suggested.
The discussion needs to be rewritten; there are too many technical elements proper to the results. The results, i.e., quantitative data, species lists, factual observations, need to be integrated into the results part. The discussion should focus on the study as a whole, highlighting limitations and qualities, and discussing perspectives. It is also important to integrate here the elements of significance of the study as highlighted above (*).
Response: we have rewritten the discussion section, as suggested, and added the significance, limitation, and perspective of this study in the manuscript: “Significance, Limitation, and Perspective. Despite BPA is rapidly metabolized to BPA conjugates in humans and then rapidly excreted, but this does not necessary mean the negligible risk of BPA exposure, since BPA metabolites are still widely present in human blood (Ginsberg and Rice 2009). Moreover, BPA metabolites have been proved to have toxic effects on humans (Ginsberg and Rice 2009, Safe 2000, Volkel et al. 2002). Therefore, it is important to understand the underlying mechanisms responsible for the metabolism of BPA in humans. This study first demonstrates that continuous BPA ex-posure disrupted the rat gut microbiota community, which in turn altered the metabolism of BPA in rat. Rat gut microbiota significantly correlated with the profile of BPA metab-olites in rat were also identified. This finding contributes to the better understanding of metabolism of BPA in humans. Notably, gut microbiota is highly variable between dif-ferent species. Thus, the transposition of the results of this study to humans should be performed with caution. More studies are needed to further evaluate the relative con-tribution of gut microbiota to the occurrence of BPA metabolites in humans.”

Reviewer 3 Report
My compliments to the authors for the research they have carried out. It is, in my opinion, an important piece in the knowledge of the entire problem of microplastics and human health.
I have no particular indications to report except for two details:
(1) on line 159 there is a Celsius degree symbol that looks superscripted and not normal, but MOSTLY...
(2) the references are not indicated as required by the instructions for the Authors which I reproduce below:
References must be numbered in order of appearance in the text (including table captions and figure legends) and listed individually at the end of the manuscript. We recommend preparing the references with a bibliography software package, such as EndNote, ReferenceManager or Zotero to avoid typing mistakes and duplicated references. We encourage citations to data, computer code and other citable research material. If available online, you may use reference style 9. below.
Citations and References in Supplementary files are permitted provided that they also appear in the main text and in the reference list.
In the text, reference numbers should be placed in square brackets [ ], and placed before the punctuation; for example [1], [1–3] or [1,3]. For embedded citations in the text with pagination, use both parentheses and brackets to indicate the reference number and page numbers; for example [5] (p. 10). or [6] (pp. 101–105).
Author Response
My compliments to the authors for the research they have carried out. It is, in my opinion, an important piece in the knowledge of the entire problem of microplastics and human health.
I have no particular indications to report except for two details:
Response: we thank the reviewer very much for your comments.
(1) on line 159 there is a Celsius degree symbol that looks superscripted and not normal, but MOSTLY...
Response: we have revised that issue.
(2) the references are not indicated as required by the instructions for the Authors which I reproduce below:
References must be numbered in order of appearance in the text (including table captions and figure legends) and listed individually at the end of the manuscript. We recommend preparing the references with a bibliography software package, such as EndNote, ReferenceManager or Zotero to avoid typing mistakes and duplicated references. We encourage citations to data, computer code and other citable research material. If available online, you may use reference style 9. below.
Citations and References in Supplementary files are permitted provided that they also appear in the main text and in the reference list.
In the text, reference numbers should be placed in square brackets [ ], and placed before the punctuation; for example [1], [1–3] or [1,3]. For embedded citations in the text with pagination, use both parentheses and brackets to indicate the reference number and page numbers; for example [5] (p. 10). or [6] (pp. 101–105).
Response: we thank the reviewer very much for your instruction. We will revise all of the reference, after reviewing by the editors of TOXICS.